# Passive motion of the lower extremities in sedated and ventilated patients in the ICU – a systematic review of early effects and replicability of Interventions

Rahel Vollenweider[1], Anastasios I. Manettas[1], Nathalie Häni[1], Eling D. de Bruin[2,3,4]*, Ruud H. Knols[2,5]

1 Nursing and Allied Health Profession Office, Physiotherapy Occupational Therapy, University Hospital Zurich, Zurich, Switzerland, 2 Department of Health Sciences and Technology, Institute of Human Movement Sciences and Sport, ETH Zurich, Zurich, Switzerland, 3 Division of Physiotherapy, Department of Neurobiology, Care Sciences and Society, Karolinska Institutet, Solna, Sweden, 4 OST–Eastern Swiss University of Applied Sciences, Department of Health, St. Gallen, Switzerland, 5 Directorate of Research and Education, Physiotherapy Occupational Therapy Research Center, University Hospital Zurich, Zurich, Switzerland

* eling.debruin@hest.ethz.ch

**Data Availability Statement:** All relevant data are within the manuscript and its Supporting Information files.

## Abstract

Early mobilization, which includes active / passive motion in bed along with mobilization out of bed, is recommended to prevent the development of intensive care unit acquired-weakness (ICU-AW) for patients with critical illness on the intensive care unit. To date, the impact of passive motion of the lower extremities in sedated and ventilated patients remains unclear. The aim of the study is to systematically review and summarize the currently available randomized controlled trials in English or German language on the impact of passive motion of the lower extremities in sedated and ventilated patients $\geq$ 18 years in the intensive care unit on musculature, inflammation and immune system and the development of intensive care unit-acquired weakness and to evaluate the replicability of interventions and the methodological quality of included studies. A systematic literature search was performed up to 20th February 2022 in the databases Medline, Embase, Cochrane Library, CINAHL and PEDro. The description of the intervention (TIDieR checklist) and the methodological quality (Downs and Black checklist) were assessed. Five studies were included in the qualitative syntheses. On average, the studies were rated with 6.8 out of 12 points according to the TIDieR checklist. For the methodological quality an average of 19.8 out of 27 points on the Downs and Black checklist was reported. The results of included studies indicated that muscle loss may be reduced by passive manual movement, passive cycling and passive motion on a continuous passive motion-unit. In addition, positive effects were reported on the reduction of nitrosative stress and the immune response. The impact on the development of ICU-AW remains unclear. In conclusion, passive movement show a slight tendency for beneficial changes on cellular level in sedated and ventilated patients in the ICU within the first days of admission, which may indicate a reduction of muscle wasting and could prevent the development of ICU-AW. Future randomized controlled trials should use larger samples, use complete

**Funding:** The authors received no specific funding for this work.

**Competing interests:** The authors have declared that no competing interests exist.

**Abbreviations:** APACHE II, Acute Physiology And Chronic Health Evaluation II; CG, Control Group; CPM, Continuous Passive Motion; F/E, Flexion and Extension; F, Female; FITT, Frequency, Intensity, Time, Type of exercise; $HbO_2$, Oxyhemoglobin and Oxymyoglobin; HHb, Deoxyhemoglobin and Deoxymyoglobin; IG, Intervention Group; ICU, Intensive Care Unit; ICU-AW, Intensive Care Unit-Acquired Weakness; IFN-γ, Interferon-γ; IL-6, Interleukin-6; IL-10, Interleukin-10; LE, Lower Extremity; M, Male; M., Musculus; MeSH, Medical Subject Headings; MRC, Medical Research Scale; NIRS, Near-Infrared Spectroscopy; no., Number of patients; NO, Nitric Oxide; PF/DE, Plantar Flexion/ Dorsal Extension; PT, Physical Therapist; QFT, Quadriceps Femoris Thickness; RASS, Richmond Agitation Sedation Scale; rep., repetitions; rev/min, revolutions per minute; SAPS, Simplified Acute Physiology Score; SMD, Standardized Mean Difference; THb, Total Hemoglobin; TNF-α, Tumor Necrosis Factor-alpha.

intervention description, use a comparable set of outcome measures, use rigorous methodology and examine the effect of passive motion on the development of ICU-AW.

# Introduction

Due to the unstable condition at the beginning of their disease, critically ill patients often require mechanical ventilation and analgosedation. In consequence, immobility occurs, which leads to muscle degradation up to 30% already within ten days of inactivity, which is primarily reflected in a reduced size of muscle fibers [1, 2]. In addition, the patients often suffer from systemic inflammation because of shock, trauma, sepsis or due to the critical illness itself and it is known that in this context pro-inflammatory cytokines increase the degradation of muscle proteins [3].

Both immobility and inflammation are among the most frequently mentioned risk factors that, in combination with the existing severe illness, favor the "intensive care unit-acquired weakness" (ICU-AW) [4]. This complication affects up to 67% of patients who have been ventilated for more than ten days and 80% of critically ill patients [4–6]. ICU-AW clinically manifests as symmetrical flaccid paresis of the extremities and not only prolongs the stay in the intensive care unit (ICU), but is also associated with increased morbidity and mortality [7]. Furthermore, ICU-AW, although reversible in principle, often leads to disability that lasts until after the acute hospital [3, 5]. Especially the strength of the lower extremities often remains permanently limited, which negatively affects the quality of life after surviving a severe illness due to the reduced ability to walk [8, 9].

It is reported that the concept of early mobilization (active and passive, in and out of bed) leads to decreased incidence of ICU-AW, shorter delirium duration, more ventilator-free days, improved muscle strength, decreased muscle atrophy and length of hospital stay, a better functional outcome at hospital discharge and increased discharged-to-home rate for patients with a critical illness [10–15].

Due to cardiac, haemodynamic or pulmonary instability and the resulting need for ventilation and analgosedation, it is usually not yet possible for patients to actively cooperate and getting mobilized in a chair during the first few days in the ICU. However, early mobilization of critically ill patients in the ICU is recommended within 48 to 72 hours after the start of mechanical ventilation, whenever possible [1, 16].

Passive movements and bed cycling are recommended in sedated patients with a RASS (Richmond Agitation Sedation Scale) ≥ -3 [17] and as early as possible in the course of treatment [18]. Despite the widespread use of the recommendations for passive movements and bed cycling and their reported safety and feasibility [19–22], these interventions lack a firm evidence-base concerning the effectiveness of passive motion treatment methods in bedridden patients [23–25], notwithstanding this patient population is at greatest risk of developing ICU-AW [24].

One study result indicates continuous passive motion (CPM) to significantly reduce muscle fibre atrophy and protein loss, when it is compared to standard therapy [26]. Another study reports passive cycling to result in significantly improved muscle strength after the intervention [27]. However, whether the passive movements can prevent muscle tissue from atrophy remains unclear in both studies [26, 27].

Data from other studies suggest passive bed cycling to improve anti-inflammatory processes and the immune response in critically ill patients and thus could prevent muscle degradation and the development of ICU-AW [28].

Yet, what is unclear is whether passive early motion measures of the lower extremities used in sedated and ventilated ICU patients reduces muscle wasting, has positive effects on inflammation and the immune system and could, therefore, prevent the development of ICU-AW.

For the translation of scientific findings from positive study results into everyday therapy it is essential that intervention descriptions are complete and replicable [29, 30]. For detailed descriptions of interventions, the TIDieR checklist was proposed and used in several publications [29, 31, 32].

This systematic review aims to summarize the effects of passive motion of the lower extremities in sedated and ventilated patients in the ICU on musculature, inflammation, immune system and the development of ICU-AW. Furthermore, the aim was to evaluate the replicability of the used interventions and the methodology of the included studies for practical settings.

## Materials and methods

A systematic review was conducted for which the methodological procedure was based on the PRISMA guideline (see S1 and S2 Checklists) [33]. The systematic review was not registered.

### Information sources and search

The terms population, intervention and study design were used to develop the search strategy (Table 1). The search strategy contained MeSH terms (critical illness, intensive care units, bed rest / immobilization, humans / adult, deep (conscious) sedation, respiration, artificial, motion therapy, continuous passive, exercise therapy, lower extremity, physical therapy specialty / modalities, rehabilitation, movement, bicycling) and free text words (critically ill, sedat*, mechanical ventilat*, early passive exerc* / cycl* / mobile* / «range of motion» / therap* / treat* / train*, leg / limb, bed exercise, recumbent / in bed, kinetic therapy). In addition, the search strategy was restricted to randomized controlled trials.

The electronic literature search was conducted by a professional librarian at the University of Zurich (SK), in the databases Medline, Embase, Cochrane Library, CINAHL and PEDro up to 6th May 2020. The complete search strategy is stated in S2 Checklist. To check whether further relevant RCT studies were published after May 2020, an additional search of the databases up to 20th February 2022 was performed.

### Eligibility criteria

Studies were included in this review if they were 1) randomized controlled trials 2) in English or German language, 3) included mechanically and invasively ventilated and sedated critically ill patients $\geq$ 18 years, who require treatment in the ICU in an acute hospital, 4) evaluated the effect of passive motion of the lower extremities carried out in bed, either manually or through a therapy device, on musculature, inflammation, the immune system or the development of ICU-AW and 5) included a comparison group that received either no therapy, standard therapy or a different dosage of the intervention.

Passive motion is used as a generic term for all passive measures that counteract the negative consequences of immobility in patients requiring intensive care. The measure is carried out manually or by means of a therapy device, is performed within the bed and results in movement without the active cooperation of the patient. As long as the passive measures take place in the ICU, they are counted as "early" at any time—i.e. even after 72h after admission to the ICU. Sedated patients were defined as deeply sedated to non-awakable (RASS -4 to -5, Ramsay Score > 4). Standard therapy was defined as respiratory therapy or nursing measures and positioning.

**Table 1. Search strategy.**

| | Description | MeSH-Terms | Free Text Words |
|---|---|---|---|
| **Population** | Ventilated and sedated critically ill patients in the intensive care unit | • Critical illness<br>• Intensive care units<br>• Bed rest / immobilization<br>• Humans / adult<br>• Deep (Conscious) sedation<br>• Respiration, Artificial | • Critically ill<br>• Sedat*<br>• Mechanical ventilat* |
| **Intervention** | Passive motion of the lower extremities manually or by a therapy device | • Motion therapy, continuous passive<br>• Exercise therapy<br>• Lower extremity<br>• Physical therapy specialty/ modalities<br>• Rehabilitation<br>• Movement<br>• sBicycling | • Early passive exerc* / cycl* / mobili* / motion / «range of motion» / therap* / treat* / train*<br>• Leg / Limb<br>• Bed exercise<br>• Recumbent / in bed<br>• Kinetic therapy |
| **Control** | no therapy, standard therapy or other dosage of the measure | - | - |
| **Outcome** | Musculature<br>Inflammation /<br>Immune system<br>Development of an ICU-AW | • Muscle strength / physiology<br>• Inflammation<br>• Cytokines<br>• Interleukins / Tumor Necrosis Factor-alpha | • Outcome<br>• Muscle cross-section / biopsy<br>• Electrophysiology Testing<br>• Blood analysis<br>• Adverse effects<br>• feasib*<br>• Medical Research Council Scale (MRC)<br>• Medical Research Council Scale Sum Score (MRC-SS)<br>• Intensive care unit-acquired weakness (ICU-AW) |
| **Design** | RCT | Randomized Controlled Trials | - |

Presentation of the search terms for the literature search following the PICOS scheme [34].

Legend: MeSh-Terms = Medical Subject Headings, RCT = randomized controlled trial.

Studies with children, with animals or those written in another language were excluded. In addition, studies with passive early motion in combination with other early rehabilitation measures, such as active early motion, electrotherapy or mobilization out of bed, were excluded.

## Study selection

After conducting the literature search, duplicates were screened out by SK. The screening of the studies was carried out in several steps by two independent reviewers (NH, RV) using the defined inclusion and exclusion criteria. In case no agreement could be reached between these two reviewers, RK acted as an independent referee.

In a first step, NH and RV performed a title and abstract screening to remove obviously irrelevant references. The refences were therefore marked as "clearly include", "maybe include" or "clearly exclude". The second step involved full-text screening with the references that remained. Finally, the third step involved a hand search by RV reviewing the reference lists of the included studies. In addition, RV contacted authors to ask for missing information that would allow inclusion of the studies.

## Data collection process and data items

Following the Cochrane checklist, the first author extracted data on 1) reference with publication year, 2) study design, 3) population, 4) setting, 5) intervention, 6) control group, 7) outcome variables and 8) results [35]. To describe the population age, gender, duration of ventilation at the start of the intervention, sedation depth, severity of illness and reason for admission to the ICU were extracted. The presentation of the measures implemented in the intervention and control group is based on the reporting of Frequency, Intensity, Time, and Type (FITT) intervention components [36]. The measurement methods and the timing of the measurements are presented for the relevant target variables. In addition, the group differences, the significance level and the effect size are described for the results.

In case the corresponding data were not presented in an included study, RV contacted the authors concerned in order to be able to include the data in this review.

## Description of the intervention (TIDieR checklist)

The description of the intervention was assessed by two independent investigators (RV and AM). The items of the TIDieR Checklist were rated as either "sufficient" (+) or as "not / not adequately" reported (-).

In the event of disagreements between AM and RV, RK was called in as referee.

## Risk of bias in individual studies

The methodological quality of the randomized controlled trials was assessed by two independent reviewers (AM and RV) using the Downs and Black checklist [37]. It was agreed that 0 points (for "no" or "unclear") or 1 point (for "yes") would be awarded for each item. For item 4, one point is only awarded in case all FITT criteria were described [36]. When no information was described for item 19 (assessment of compliance with the intervention), this was rated as "yes" in the ICU setting and 1 point was awarded. Items 9 and 26 were scored as "yes" and 1 point when the loss-to-follow-up rate was reported and was < 15%. Finally, for Item 27, 1 point was awarded when a sample size calculation was performed and 0 points when this item was missing. These adjustments lead to a total possible score of 27 points.

In the event of disagreements between AM and RV, RK was called in as referee.

## Summery measures

To show the effect of the interventions, the standardized mean difference (SMD), the so-called effect size, was calculated for the randomized controlled trials that reported the mean and standard deviation of the relevant outcome variables. The effect size was determined by calculating Cohen's *d*, whereby a *d* between 0.2 and 0.5 indicates a small effect, between 0.5 and 0.8 a medium effect, and a *d* greater than 0.8 indicates a strong effect [38].

In order to show the agreement of the reviewers for the assessment of the risk of bias with the Downs and Black checklist, the Cohen's Kappa coefficient was calculated [39]. A kappa value below 0.20 is considered a weak agreement, between 0.21 and 0.40 a slight agreement, between 0.41 and 0.60 a moderate agreement, between 0.61 and 0.80 a good agreement, and greater than 0.81 a very good agreement [40].

## Synthesis of results

A meta-analysis was intended to be provided when sufficient data would be available.

## Results

### Study selection

The literature search (search strategy S2 Checklist) resulted in 2810 hits. After removing the duplicates, the number of references was reduced to 1817. After the additional search, no studies could be identified.

The Flow Chart in Fig 1 show the process of study selection based on the predefined inclusion and exclusion criteria. After the title and abstract screening, 1740 studies could be excluded. The remaining 77 studies were read in full text. This full text screening led to the exclusion of a further 72 studies due to inappropriate population, intervention or control group, different language or because they were not clinical trials. No further study could be included by hand-searching of the references of the selected manuscripts. However, contacting three authors resulted in further study inclusion. Finally, five studies were integrated into the qualitative synthesis.

A search of the databases between May 2020 and February 2021 did not result in additional references.

### Study characteristics

The characteristics of the five included studies [24, 26, 41–43] are presented in Table 2. All studies were conducted in intensive care units, three in Brazil, one in the United Kingdom, and one in France.

The total of 87 patients (19 women and 49 men, 19 patients' sex unclear) were on average 56.4 years old. The number of days with ventilation before the start of the study was only stated in two studies and ranged from a median of 4 days [24] to 6.44 ± 0.333 days in the intervention group and 4.9 ± 2.80 days in the control group [42]. All patients in the included studies were deeply sedated (RASS -4 or -5, Ramsay score 6), comatose or relaxed. Severity of the disease was stated in three studies with the APACHE II score and showed in average 18.6 points and, with 24%, therefore a low average risk of dying in hospital [26, 42, 43]. In another study the SAPS II score was used and a value of 57.5 ± 24 suggests a 50–75% in-hospital mortality [24]. The reason for admission to the ICU was missing in two studies [41, 43]. In the other studies, respiratory failure or sepsis is most frequently mentioned [24, 26, 42].

In the included studies, three different passive early motion interventions were described for ventilated and sedated patients in the ICU. Passive exercise was performed by a therapist as a stand-alone intervention [24] or combined with the application of a cuff to restrict blood flow [41]. Passive bed cycling has been described in three studies [24, 42, 43] but each performed with a different dosage. Finally, in one study passive motion was performed by a CPM machine [26]. Apart from one study, in which the ankle joint was moved in isolation [26], the remaining studies involved movement of the entire lower extremity (hip, knee and ankle joint movement).

The outcome categories of musculature, inflammatory factors, immune system and development of ICU-AW were measured with different parameters for the same target variables in the included studies. Only muscle thickness was measured in two studies [41, 44] and also with the same measurement method. In the other studies the target variables and how these were methodologically measured differed between the reports. The target variables were measured before and then directly after the intervention [24], 1 h after the intervention [42], 7 days after the start of the intervention [26, 43] or as soon as the patients were no longer sedated and could respond to requests [41].

Effects of passive interventions with group differences, p-value, and calculated effect sizes are reported in Table 2.

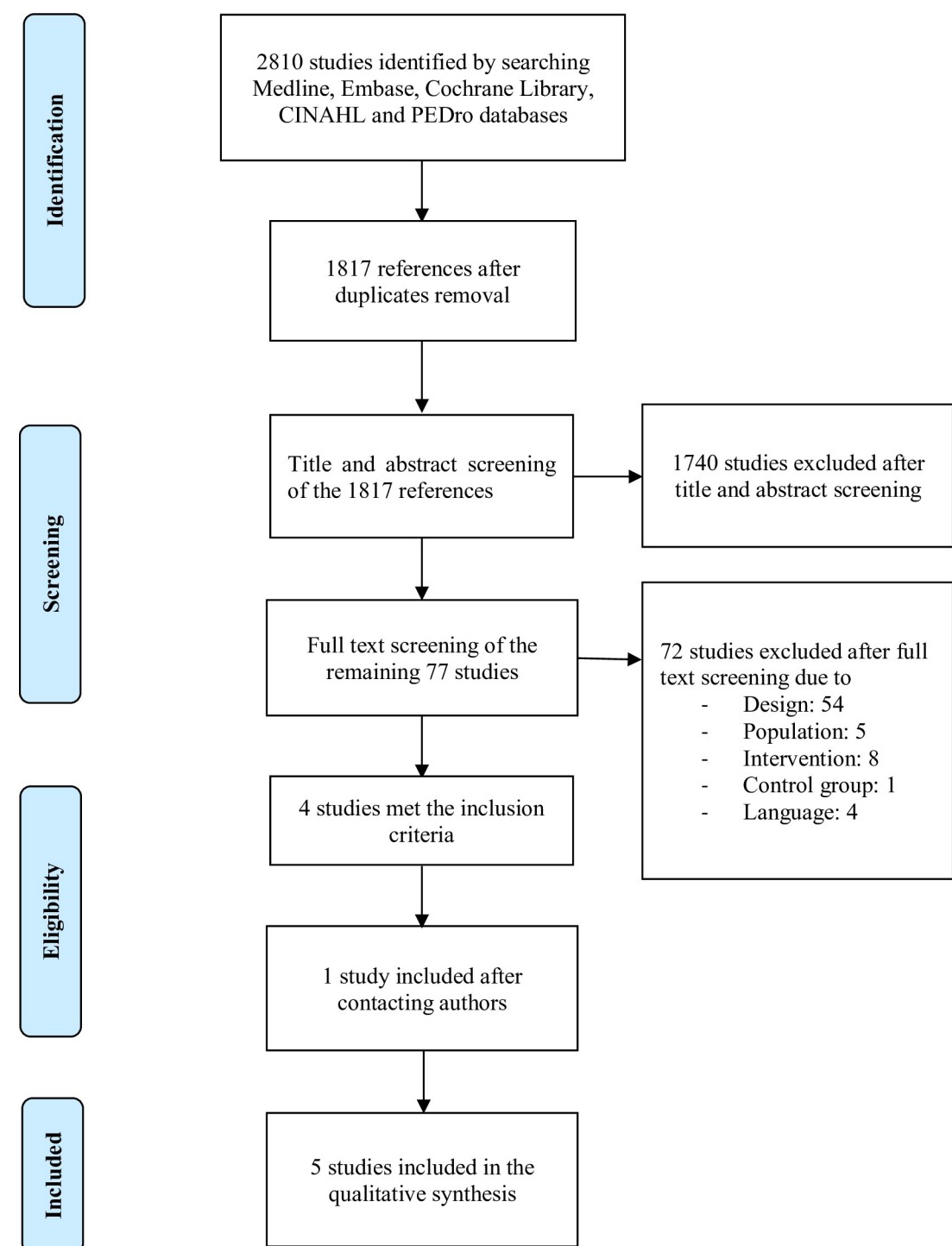

**Fig 1. Flow chart.** Presentation of the study selection process [33].

### Assessment of intervention description

The assessment of intervention description was conducted independently by two reviewers (AM and RV) using the TIDieR checklist [29]. An overview is presented in Table 3.

**Table 2. Characteristics of included studies.**

| Reference | Design | Population | Setting | Intervention | Control | Outcome | Results |
|---|---|---|---|---|---|---|---|
| **Barbalho (2019)** | Within-patient randomized trial | No. = 20 m: 17, f: 3, 66 ± 4.3 yrs. old, comatose patients IG: no. = 20, CG: no. = 20 Reason for entry on ICU unclear | Two ICU in a reference hospital in Belém (Brasil) | Application of a cuff in the proximal part of one leg to restrict blood flow with a pressure value of 80% of the pat.'s systolic blood pressure in the anterior tibial artery. Passive motion with the cuff during the ICU stay, 1x/day, 3x15 repetitions F/E in the knee joint with 2s duration for flexion and extension. | Passive motion without the cuff during the ICU stay, 1x/day, 3x15 repetitions F/E in the knee joint with 2s duration for flexion and extension. | Muscle circumference and thickness (measured with ultrasound) of the quadriceps femoris muscle before the start of the intervention and after the end of the intervention without sedation and with the presence of volitional motor function. Muscle strength (MRC sum score and lower extremity separately) was measured once the patient was no longer sedated and could respond to requests. | The intervention was performed 11 ± 2.2 days. Muscle circumference and thickness were significantly reduced in both the intervention and control groups (p = 0.001). However, application of a cuff to the upper leg with passive exercise resulted in a 6.5% lower muscle loss (p = 0.001) compared to passive exercise alone (muscle circumference I: $d = 0.86$, C: $d = 1.40$, muscle thickness I: $d = 0.80$, C: $d = 1.03$). MRC sum score was 50.9 ± 2.78 ($< 48$ = clinical diagnosis of ICU-AW) and lower extremity score was 24 ± 3. |
| **Franca (2020)** | Controlled randomized open clinical trial | No. = 19 IG: no. = 9, 60.11 ± 27.15 yrs. old, RASS–4.89 ± 0.333, APACHE II score 22.67 ± 8.36, 6.44 ± 0.333 days ventilated at start of intervention CG: no. = 10, 56.80 ± 12.80 yrs. old, RASS–4.50 ± 0.707, APACHE II score 22.60 ± 4.22, 4.9 ± 2.80 days ventilated at start of intervention Respiratory / cardiac disorders, infections or other reason for entry on ICU | 16 bed ICU in the Agamenon Magalhães Hospital (Brasil) | Passive bed cycling of the lower extremities during 20min at a speed of 30 revolutions/min, single performance of the intervention. | No intervention | Analysis of nitrosative stress by determination of nitric oxide (NO) production in monocytes and determination of inflammatory cytokines (IL-6, IL-10, TNF-α, IFN-y) directly before and 1h after the intervention | 1 h after the intervention, there was a significantly reduced nitric oxide concentration (NO (C+), NO (CO-)) in the blood of the intervention group (p < 0.001). TNF-α was also significantly reduced by the intervention (p = 0.049). In the group comparison, the intervention showed a small effect on the reduction of nitric oxide concentration compared to the control group (d = 0.426, d = 0.416, p-value n.a.). Among the cytokines, there was a small effect on the reduction of IL-6 (d = -0.347) and a large effect of the intervention compared to the control group on the increase of IL-10 (d = 2.885). In contrast, the intervention increased IFN-γ (d = 0.797) and there was no effect of the intervention compared to the control group on TNF-α (d = 0.195). |

(*Continued*)

**Table 2.** (Continued)

| Reference | Design | Population | Setting | Intervention | Control | Outcome | Results |
|---|---|---|---|---|---|---|---|
| **Griffiths (1995)** | Within-patient randomized clinical trial | no. = 5 m: 3, f: 2, 44.80 ± 16.02 yrs. old, relaxed patients, APACHE II score at entry on ICU: 17.2 ± 4.92, maximal: 20.4 ± 7.83 IG: 5, CG: 5 Reason for entry on ICU: Pat. 1: abdominal sepsis with respiratory failure. Pat. 2: Rib fractures, sepsis, pneumonia and renal failure Pat. 3: Flail chest, pulmonary and hepatic contusion, sepsis after traffic accident as pedestrian Pat. 4: Flail chest, pulmonary contusion after traffic accident as a car driver Pat. 5: Flail chest after fall with osteogenesis imperfecta | ICU at Whiston Hospital in Prescot (United Kingdom) | Passive movement in the ankle joint of one leg in PF/DE on a Straumann splint (CPM splint) during 7 days, 3x daily for 3h. Splint setting: knee joint in rest position of 140 ± 5° flexion, movement of ankle in PF/DE with ROM of 25% of the functional passive stretching capacity of the tibialis anterior muscle in addition to standard therapy with 2x/day < 5min passive stretching of both lower extremities by a PT | Standard therapy of the other leg with 2x/day < 5min passive stretching of the lower extremities by a PT | Percutaneous muscle biopsy of the anterior tibialis muscle on both sides for analysis of muscle tissue immediately before and at the end of the intervention after 7 days: • Muscle cross section • muscle fiber types • muscle biochemistry: RNA, DNA and protein levels • Polyribosome profile (representation of ribosomes involved in active protein synthesis) | The intervention showed significant changes only in the more severely ill patients (APACHE II score > 19). Intervention in addition to standard therapy significantly reduced muscle fiber atrophy in severely ill patents (no. = 3, p = 0.025) compared to standard therapy alone. Type I muscle fibers were preserved significantly more pronounced than type II fibers fibers (n = 3, p = 0.02). Muscle loss was significantly higher in the control leg (p = 0.012). Protein content also decreased in both legs, but was significantly greater in the control leg (p = 0.04). Effect sizes could not be calculated because of missing data |
| **Medrinal (2018)** | Randomized controlled cross-over trial | no. = 19 m: 13, f: 6, 65.3 ± 9.7yrs. old, Ramsay Score: 6, ventilated for a median of 4 days at the start of intervention, SAPS-II-Score: 57.5 ± 4 IG: 19, CG: 19 Reason for entry on ICU: Pneumonia, sepsis, COPD/asthma, exacerbation, heart failure, substance use/acute mental status, intra-abdominal sepsis with surgery | 16 beds ICU in the ICU department of the Le Havre Hospital Group (France) | Single procedure, patients received all interventions with 30min rest in between I[1]: 10min passive knee joint flexion and extension and hip joint abduction and adduction I[2]: 10min passive bed cycling at a speed of 20 rev/min | Patients act as their own control group after a 30-minute break | Muscle microcirculation by NIRS measurement in the quadriceps femoris vastus lateralis muscle before and after the 10-minute interventions. The relative change in total hemoglobin (THb), oxyhemoglobin and oxymyoglobin (HbO$_2$), and deoxyhemoglobin and deoxymyoglobin (HHb) was measured. | Passive exercise significantly reduced THb by 23% (p = 0.046) and HHb by 27% (p < 0.05). The intervention had no effect on HbO$_2$. Compared to the control group, the intervention had a small effect on the reduction of THb (d = 0.221) and HbO$_2$ (d = 0.319) of the quadriceps vastus lateralis muscle. No effect is evident for HHb (d = 0.117). Passive bed cycling increased THb (+12.5%) and HbO$_2$ (+11%), but not significantly. In the group comparison, the intervention showed no effect on the microcirculation of the quadriceps vastus lateralis muscle compared to the values after a 30-minute break (THb: d = 0, HbO$_2$: d = 0.172, HHb: d = 0.04). |

(*Continued*)

**Table 2.** (Continued)

| Reference | Design | Population | Setting | Intervention | Control | Outcome | Results |
|---|---|---|---|---|---|---|---|
| **Ximenes Carvalho (2019)** | Controlled randomized pilot study | No. = 24 m: 16, f: 8, RASS –4 IG: n = 12, 47.83 ± 19.61 yrs. old, APACHE II score: 14.42 ± 6.25 CG: no. = 12, 54.17 ± 16.71 yrs. old, APACHE II score: 16.00 ± 5.84 | ICU of the University Hospital of Santa Maria (Brasil) | Passive bed cycling of the lower extremities with 30° upper body elevation in bed during the first week (7 days) in the ICU, 1x/day during 20min at a speed of 20 revolutions/min in addition to standard therapy | Standard therapy with respiratory therapy and motor therapy during 7 days 2x/day 30min (includes the following measures: Vibrocompression, hyperinflation by ventilator and if necessary tracheal suctioning, passive and active-assisted motion therapy of upper and lower extremities in bed) | Quadriceps femoris thickness (QFT) measured by ultrasound within 48h with mechanical ventilation and after 7 days (at the end of the protocol) | Muscle thickness remained unchanged in both groups after implementing the protocol (Control: left QFT p = 0.558; right QFT p = 0.682, Intervention: left QFT p = 0.299; right QFT p = 0.381) Comparing the groups, there was a small effect of the intervention compared to the control group on the preservation of muscle thickness of the quadriceps femoris (UE right: d = 0.318, p = 0.738, UE left: d = 0.381, p = 0.248), but this was not significant. |

Presentation of characteristics of included studies based on the Cochrane Checklist [35].

Studies are listed alphabetically in the table, and only those data from the studies that are relevant to answering the research question are presented.

Legend: APACHE II score = Acute Physiology And Chronic Health Evaluation II score (classification system to calculates risk of dying in hospital, the higher the calculated value, the higher the risk of dying in hospital (minimum value = 0; Maximum value = 71) [45]), CG = control group, CPM = continuous passive motion, F/E = flexion and extension, f = female, $HbO_2$ = oxyhemoglobin and oxymyoglobin, HHb = deoxyhemoglobin and deoxymyoglobin, IG = intervention group, ICU = intensive care unit, ICU-AW = intensive care unit-acquired weakness, IFN-γ = interferon γ, IL-6 = interleukin-6, IL-10 = interleukin-10, LE = lower extremity, m = male, M. = musculus, MRC = Medical Research Scale (muscle function testing), NIRS = near-infrared spectroscopy, no. = number of patients, NO = nitric oxide (NO (C-) = NO production of non-stimulated monocytes, NO (C+) = NO production of stimulated monocytes), PF/DE = plantar flexion/dorsal extension, PT = physical therapist, QFT = Quadriceps femoris thickness, RASS = Richmond Agitation Sedation Scale, rep. = repetitions, rev/min = revolutions per minute, SAPS = Simplified Acute Physiology Score (assesses mortality risk of intensive care patients on admission to the ICU, the higher the calculated score, the higher the in-hospital mortality is estimated to be [46]), THb = total hemoglobin, TNF-α = tumor necrosis factor-alpha.

On average, the studies were rated with 6.8 points, with a range of 6–8 points being awarded for the individual studies. The name of the intervention, the underlying principles, the materials used, the procedure and the dosage of the intervention (items 1–4 and 8) were best reported and could be rated as sufficiently reported (+) in all studies. Items 6 (modality of delivery) and 10–12 (modifications and treatment adherence) were not reported or not reported adequately in any study and were accordingly rated as (-).

**Table 3. Evaluation of the intervention description.**

| Referenz \ Item | 1 | 2 | 3 | 4 | 5 | 6 | 7 | 8 | 9 | 10 | 11 | 12 | Total «+» per study |
|---|---|---|---|---|---|---|---|---|---|---|---|---|---|
| Barbalho (2019) | + | + | + | + | - | - | + | + | + | - | - | - | 7 |
| Franca (2020) | + | + | + | + | - | - | + | + | + | - | - | - | 7 |
| Griffiths (1995) | + | + | + | + | - | - | - | + | + | - | - | - | 6 |
| Medrinal (2018) | + | + | + | + | + | - | + | + | + | - | - | - | 8 |
| Ximenes Carvalho (2019) | + | + | + | + | - | - | + | + | - | - | - | - | 6 |
| Total «+» per Item | 5 | 5 | 5 | 5 | 1 | 0 | 4 | 5 | 4 | 0 | 0 | 0 | |

Illustration of the assessment of intervention reporting using the TIDieR checklist [29].

Legend: + = sufficiently reported,— = not or not sufficiently reported.

## Risk of bias

The included studies achieved an average score of 19.8, with a median of 22 points on the Downs and Black Checklist [37]. An overview is presented in Table 4. Three studies scored the highest with 22 points [24, 41, 43] and one scored the lowest with 13 points [26].

The total per item reported ranged from 0 to 5 points with a median of 4 points. In all studies, full scores were obtained on twelve items (2, 3, 5–7, 10, 14, 16, 17, 19, 22, 23). In contrast, item 12 ("Were those subjects who were prepared to participate representative of the entire population from which they were recruited?" [37] and item 25 ("Was there adequate adjustment for confounding in the analyses from which the main findings were drawn?" [37]) was rated with 0 points in all studies.

**Table 4. Risk of bias in individual studies.**

| Reference / Item | Barbalho (2019) | Franca (2020) | Griffiths (1995) | Medrinal (2018) | Ximenes Carvalho (2019) | Total per item |
|---|---|---|---|---|---|---|
| 1 | 1 | 1 | 0 | 1 | 1 | 4 |
| 2 | 1 | 1 | 1 | 1 | 1 | 5 |
| 3 | 1 | 1 | 1 | 1 | 1 | 5 |
| 4 | 1 | 1 | 0 | 1 | 1 | 4 |
| 5 | 1 | 1 | 1 | 1 | 1 | 5 |
| 6 | 1 | 1 | 1 | 1 | 1 | 5 |
| 7 | 1 | 1 | 1 | 1 | 1 | 5 |
| 8 | 0 | 0 | 0 | 1 | 1 | 2 |
| 9 | 1 | 0 | 0 | 1 | 0 | 2 |
| 10 | 1 | 1 | 1 | 1 | 1 | 5 |
| 11 | 1 | 1 | 0 | 0 | 1 | 3 |
| 12 | 0 | 0 | 0 | 0 | 0 | 0 |
| 13 | 1 | 1 | 0 | 1 | 1 | 4 |
| 14 | 1 | 1 | 1 | 1 | 1 | 5 |
| 15 | 0 | 0 | 0 | 0 | 1 | 1 |
| 16 | 1 | 1 | 1 | 1 | 1 | 5 |
| 17 | 1 | 1 | 1 | 1 | 1 | 5 |
| 18 | 1 | 1 | 0 | 0 | 1 | 3 |
| 19 | 1 | 1 | 1 | 1 | 1 | 5 |
| 20 | 1 | 1 | 1 | 1 | 1 | 5 |
| 21 | 1 | 1 | 0 | 1 | 1 | 4 |
| 22 | 1 | 1 | 1 | 1 | 1 | 5 |
| 23 | 1 | 1 | 1 | 1 | 1 | 5 |
| 24 | 1 | 0 | 0 | 1 | 1 | 3 |
| 25 | 0 | 0 | 0 | 0 | 0 | 0 |
| 26 | 1 | 0 | 0 | 1 | 0 | 2 |
| 27 | 0 | 1 | 0 | 1 | 0 | 2 |
| **Total per study** | **22** | **20** | **13** | **22** | **22** | |

Presentation of the assessment of the risk of bias of the individual studies using Downs and Black checklist [37].

The studies were sorted alphabetically. The Down and Black checklist was adapted as follows: 0 points (for "no" or "unclear") or 1 point (for "yes") are awarded for each item. For item 4, one point is only awarded if all FITT criteria (F = Frequency, I = Intensity, T = Time, T = Type of exercise) are described [36]. If no information is described for item 19 (assessment of compliance for the intervention), this will be scored as "yes" in the ICU setting and 1 point will be awarded. Items 9 and 26 are scored as "yes" and 1 point if the loss-to-follow-up rate was reported and is < 15%. Finally, item 27 is scored 1 point if a sample-size calculation was performed and 0 points if it was missing. These adjustments result in a total score of 27 points.

## Agreement of the reviewers

The agreement of the two investigators (RV and AM) in assessing the methodological quality of the studies with the Downs and Black checklist was calculated using the Kappa coefficient (Table 5). It was found that only slight agreement was obtained after the independent analysis of the studies. However, in the joint discussion, agreement could finally be reached on all items without consulting RK.

## Effects of early passive motion

In Table 2 the described effects of passive early motion of sedated and ventilated patients in the ICU are presented. Both the extracted data and the standardized mean differences calculated are presented. Standardized mean differences were calculated for the randomized controlled trials that report the mean and standard deviation of the relevant outcome variables.

**Effects on musculature.** Muscle degradation could not be prevented by passive early motion. However, passive bed cycling showed a slight effect on the preservation of muscle thickness and a slight positive tendency on the increase of microcirculation. Furthermore, the application of a cuff with additional passive exercise and the high-dose passive exercise on a CPM splint lead to significantly lesser muscle loss.

In principle, there seems to be a slight superiority of passive early motion compared to no intervention or standard therapy in terms of musculature, with a higher dosage of interventions over a longer period showing clearer results.

**Effects on inflammation and immune system.** Performing passive early motion once can reduce nitrosative stress and is also more effective than no intervention. The effect of the intervention on cytokines is unclear. On the one hand, the intervention was able to reduce the pro-inflammatory cytokines (TNF-$\alpha$, IL-6), the IL-6 even more effectively than no intervention. Furthermore, the intervention led to a more effective increase of the anti-inflammatory effective IL-10. On the other hand, an increase of the pro-inflammatory IFN-$\gamma$ is also shown. Whether these results are confirmed when the intervention is performed several times or at higher doses requires further research.

**Effects on the development of ICU-AW.** The effect of passive early motion on the development of ICU-AW could not be demonstrated in this work due to a lack of studies reporting on this outcome.

Table 5. Agreement of reviewers.

|  | A | B | Total |
|---|---|---|---|
| A | 79 | 26 | 105 |
| B | 8 | 22 | 30 |
| Total | 88 | 45 | 135 |
| $p_0$ | 0.748 | | |
| $p_e$ | 0.580 | | |
| kappa coefficient | 0.39 | | |
| **slight agreement** | | | |

Illustration of the calculation of the investigators' agreement to assess the methodological quality of the studies with the Downs and Black checklist [37] using the kappa coefficient [39].

Legend: A = Yes, B = No, $p_0$ = measured agreement value of the two estimators, $p_e$ = random expected agreement.

### Additional analysis

No quantitative syntheses were performed due to the heterogeneity of the outcome measures and insufficient methodological quality of included studies.

## Discussion

This systematic review aimed to summarize the effective passive motion measures of the lower extremities used in sedated and ventilated ICU patients to reduce muscle wasting and prevent the development of ICU-AW and to evaluate the replicability of these interventions and the methodology of the included studies. The results of this systematic review show a slight tendency of benefits related to passive motion in sedated and ventilated patients in the ICU observed in the muscle structure, in the microcirculation and in the inflammation factors and the immune system. Conversely, the efficacy for early and intensive passive movement in immobilized ICU patients could not yet be determined, as the summarized evidence was retrieved of studies with small samples. Moreover, these studies are partly replicable and their effect sizes are difficult to derive from these studies.

Then again, three of the included studies showed slight positive benefits of passive motion on muscle fibers, protein loss and muscle thickness and circumference, when compared to standard therapy or no intervention, without being able to completely prevent muscle loss [26, 41, 43].

Griffiths et al. measured significant preservation of muscle fibers and prevention of protein loss after passive movement in the ankle joint by a CPM machine in the more severely ill patients [26]. It can be considered beneficial that an effect of the intervention was evident, although only a small muscle group was passively moved and not the whole lower extremity as it was performed in the other studies [24, 41–43]. Though it must be noted that the generalizability of this study may be questioned due to the small number of patients (n = 5) and because of the deficiencies in the methodological quality. Moreover, the dosage of the intervention in this study contained a total of 9h per day, which is hardly feasible in today's daily routine in the ICU. Comparable interventions with relaxed patients were not performed since the latter publication, as relaxants for a longer time are hardly necessary nowadays with the newer ventilation technology and the existence of short-acting medications [26]. The results of significant preservation of muscle fibers are in line with the findings from the study of Llano-Diez et al. which demonstrated the effect of a high dosage of passive movement with a CPM machine [27]. In this report deeply sedated ventilated patients with brain injury of whom an extremity was passively moved by a CPM machine during 9 ± 1 days for 10 hours per day (4x 2.5 h) were observed. The study reported that the specific force of single muscle fibers in the intervention leg was 35% higher compared to the leg without passive movement. However, because this study lacked randomization the results should be interpreted with care.

The benefits observed on muscle structure through changes in microcirculation is described by the study of Barbalho et al., where muscle circumference and muscle thickness could be maintained through passive manual exercise with a blood pressure cuff [41]. This is the first study to investigate the application of a cuff to restrict blood flow in patients requiring intensive care. However, this intervention is already successfully used in musculoskeletal rehabilitation. A systematic review with blood flow restriction in addition with low-impact training showed a moderate effect on muscle strength gain compared to training alone in patients with knee osteoarthritis, ligament injuries or myositis and in older patients who are prone to sarcopenia [47]. Positive effects on the microcirculation can therefore indirectly protect the muscles, because temporary reduction in blood flow leads to a counter-reaction of the body and, in combination with exercise, even if it is low-dose, can support the maintenance or build-up of

muscle mass [41]. On the other hand, some results suggested that passive motion applied manually or by a bed cycle led to higher microcirculation, and is therefore supposed to improve blood circulation in the muscles and could reduce muscle loss [24, 41]. Whether a higher microcirculation can actually positively affect muscles needs further investigation.

Another benefit of passive motion could be seen in one of the included studies on inflammation and immune system. A single 20-minute session of passive exercise on the bed cycle showed a beneficial impact on reducing inflammation, as shown by significant reduction of nitrosative stress, measured by NO production in stimulated (C+) and non-stimulated (C-) monocytes in the blood. Furthermore, there seemed to be a positive influence on the immune system by significantly reducing the inflammatory effective TNF-α and increasing anti-inflammatory cytokine IL-10 [42]. The positive tendency of passive motion on nitrosative stress and inflammation has also been demonstrated in other studies. A study with rats described that unilateral passive mechanical loading resulted in a reduction of oxidative stress and attenuated the loss of muscle mass and force-generation capacity and could therefore show beneficial effects on muscle size and function [48]. Amidei and Sole investigated the effect of passive movement through a CPM machine on cytokines in a quasi-experimental study [28]. 30 patients with mechanical ventilation were passively moved for 20 min at a rate of 20 flexion/extension movements in the knee joint per minute within 72 hours. The patients were not all deeply sedated (GCS 3–13 at baseline), which means that active movements can distort the results. The intervention showed a significant reduction in IL-6, which indicates an anti-inflammatory effect of the intervention. IL-10 showed no significant change between baseline measurement and 60 min after the intervention. However, it is noteworthy that the ratio between IL-6 and IL-10 was also calculated in the study. This ratio could be clinically relevant, as IL-6 is a pro-inflammatory cytokine and IL-10 is an anti-inflammatory cytokine. The study showed that this ratio improved with the intervention, which certainly required a change in IL-10 concentration, but the IL-10 concentration may only be significantly reduced with a time delay.

This is one of the first systematic reviews showing benefits of a single passive intervention on cellular and structural level, although preventing muscle loss and the development of ICU-AW by counteracting the negative effects of immobility and inflammation has already been investigated in other systematic reviews.

Neuromuscular electrical stimulation (NMES) is another single passive intervention performed in the population of critically ill patients on the ICU. This intervention is controversially discussed in the literature [49–51]. On one hand two systematic reviews concluded among other things the application of NMES in ICU patients to prevent ICU-AW, maintain muscle mass and enhance their muscle strength [50, 51]. On the other hand, the systematic review with meta-analysis of Zayed et al. testified that NMES in combination with usual care was not associated with significant differences in global muscle strength [49].

Moreover, there are a lot of systematic reviews in the area of early mobilization which concentrate on various interventions and perform passive motion among other things [10, 11, 13, 52–54]. As stated in a scoping review of physical rehabilitation interventions, the most common interventions were progressive mobility and multicomponent interventions (both interventions included passive and active interventions inside and outside of the bed) [55]. Here too, the results are controversially reported regarding the effect of the combination of these interventions.

In some systematic reviews progressive mobility and multicomponent interventions appear to decrease the incidence of ICU-AW [13] and the hospital and ICU stay [11, 54], improve the functional capacity [10, 11, 13, 54] and muscle force [10, 11], increase the number of ventilator-free days [11, 13] and the discharged-to-home rate for patients with a critical illness in the

ICU setting [13]. On the other hand, in two systematic reviews the evidence regarding a benefit of early mobilization remained inconclusive [52, 53]. An explanation for this discrepancy could be the difference in timing and dosage between intervention and control group, while a bigger difference is more likely to result in a benefit [53].

Currently there are also some interventions that assess newer technologies discussed, which may have a positive impact on muscle loss and the development of ICU-AW.

Fossat et al. tried a combination of interventions and added early in-bed leg cycling and electrical stimulation of the quadriceps muscles to a standardized early rehabilitation program. However, there was no improvement in global muscle strength at discharge from the ICU [56]. Wollersheim et al. investigated the effect of muscle activation with NMES or whole-body vibration in addition to protocol-based physiotherapy on muscle strength and function. The results showed that in patients with sepsis syndrome, who are at high risk for ICU-AW, the intervention did not have a short- or long-term benefit on muscle strength and function, yet it prevented muscle atrophy [57]. Another possibility could be the combination of exercise and optimal nutrition [58, 59]. Sundström et al. reviewed the effect of active exercise and nutrition to stimulate muscle protein gain and concluded that this combination may result in reduced weakness and improved physical function. But there are only a few small RCTs showing promising results of more protein feeding on muscle thickness [58].

A typical problem of research with critically ill ICU patients are the many confounding factors that can distort the effect of the interventions.

First, the influence of medications is an important confounding factor. For example, the administration of corticosteroids and muscle relaxants increase the risk of developing ICU-AW [60, 61]. Corticosteroids also have an anti-inflammatory effect and support the immune system. In the case of increased signs of infection, patients in the ICU receive targeted anti-inflammatory drug therapy, whereby the administration of vasoconstrictive catecholamines can lead to reduced microcirculation in the case of haemodynamic instability [62]. Thus, within the included studies of this systematic review, drugs could have had a major impact on outcome parameters.

Secondly, patient nutrition must be mentioned as another confounding factor. As nutrition can have an important impact on limiting the muscle loss associated with critical illness, only with an optimal nutritional status under the prevailing conditions can its influence, for example on muscle loss, be kept as small as possible in order to demonstrate the isolated effect of an intervention [58]. The results of an ongoing study by Zhou et al. could provide information about the interaction of exercise and nutrition [63].

Thirdly, critically ill patients in intensive care often suffer from numerous comorbidities or may have installations which inhibit movements, that can have an influence on muscle loss, for example neurological diagnosis, sarcopenia, the need for extracorporeal membrane oxygenation etc. [7, 61, 64]. These factors can distort the effect of passive early motion. Clear and comparable inclusion and exclusion criteria and an accurate description of the included patients are helpful to compare the patient populations of different studies.

Apart from the confounding factors, the present work revealed that a clear definition of early mobilization and a core set of outcome variables for early mobilisation in the ICU are missing [65]. A defined set of outcome variables would allow comparability of studies. The core set should include target variables at the cellular and structural level as well as at the functional level. Furthermore, it must be possible to capture both the acute effects of the interventions and the long-term effects. On structural level it seems reasonable to measure muscle thickness with ultrasound, as this outcome measure is shown to be reliable and valid in the population of critically ill patients on the ICU [66, 67]. On the other hand, performing a muscle biopsy provides important information about the muscle fibers, but carries a higher risk of

complications due to its invasiveness and can only be performed by specialists. Furthermore, on functional level the MRC and MRC sum score can deliver information about muscle strength and the clinical diagnosis of an ICU-AW. As though a blood analysis can provide interesting information on cellular level by measuring oxidative and nitrosative stress and cytokines, these outcome measures must be performed in a specialized laboratory and are therefore more complicated to involve in everyday working life. If it is possible, especially the ratio of IL-6 and IL-10 seems to be clinically relevant [28].

In addition, studies with larger populations, longer intervention periods or higher doses, as well as the start of early mobilization within 48–72 h in the intensive care unit are lacking.

Furthermore, it would be informative to investigate whether additional passive early mobilisation also prevents muscle breakdown in ventilated patients with mild sedation and has a positive effect on inflammation, the immune system and the development of ICU-AW. This is because even in this population, only a few active interventions are often possible due to reduced exercise capacity.

It is important to investigate the effectiveness of individual interventions in high-quality studies with large populations and to describe the intervention completely in order to be able to implement positive study results in everyday clinical practice. In this way, the effective interventions can be applied in a targeted manner in intensive care units to those patients who can actually benefit from them. Especially in connection with Covid-19, effective passive early mobilisation measures could be of great importance, as the affected patients usually have to be sedated for a long time and often also have to be relaxed and show a high incidence of ICU-AW after re-awakening [68]. The results of our review show, however, that many reports lack a clear description of the interventions which hinders their full replication in a clinical setting.

## Limitations

Our systematic review has some limitations that should be mentioned. First, the included studies evaluated the effects of the interventions with different methodological measures, which prohibited to conduct a meta-analysis.

Secondly, it must be mentioned that the calculated effect sizes can be affected by biases of individual studies. Thirdly, a language bias must be assumed in the conduct of the present systematic review due to the exclusion of studies in a language other than German and English. And finally, publication bias cannot be ruled out.

## Conclusions

In conclusion, passive movement show a slight tendency for beneficial changes on cellular level in sedated and ventilated patients in the ICU within the first days of admission, which may indicate a reduction of muscle wasting and could prevent the development of ICU-AW. Future studies should possibly record both the acute and the long-term effects of the interventions. Furthermore, studies with larger populations, longer intervention periods or higher dosages, rigorous methodological quality as well as the start of early mobilisation within 48–72 h in the intensive care unit are required. These studies should preferably adhere to guidelines for reporting, e.g. the Tidier checklist, since that would allow the full replication of successful interventions in clinical settings.

## Supporting information

**S1 Checklist. PRISMA 2020 checklist.** PRISMA Checklist items in the manuscript.
(PDF)

**S2 Checklist. PRISMA 2020 for abstracts checklist.** PRISMA for Abstracts Checklist items in the manuscript.
(PDF)

**S1 File. Database search strategies.** Summary of the complete search strategy.
(PDF)

## Acknowledgments

Dr. sc. nat. Sabine Klein (SK) from the Scientific Library of the University of Zurich is acknowledged for her help in developing and conducting the search strategy of this review.

## Author Contributions

**Conceptualization:** Rahel Vollenweider, Anastasios I. Manettas, Eling D. de Bruin, Ruud H. Knols.

**Data curation:** Rahel Vollenweider.

**Formal analysis:** Rahel Vollenweider, Anastasios I. Manettas, Nathalie Häni, Ruud H. Knols.

**Investigation:** Rahel Vollenweider, Ruud H. Knols.

**Methodology:** Rahel Vollenweider, Eling D. de Bruin, Ruud H. Knols.

**Project administration:** Rahel Vollenweider, Ruud H. Knols.

**Supervision:** Eling D. de Bruin, Ruud H. Knols.

**Validation:** Rahel Vollenweider, Ruud H. Knols.

**Writing – original draft:** Rahel Vollenweider.

**Writing – review & editing:** Anastasios I. Manettas, Nathalie Häni, Eling D. de Bruin, Ruud H. Knols.

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
