## [Decision Letter · Decision Letter 0]

15 Feb 2022

PONE-D-21-16010Passive motion of the lower extremities in sedated and ventilated patients in the ICU – a systematic review of early effects and replicability of interventionsPLOS ONE

Dear Dr. de Bruin,

Thank you for submitting your manuscript to PLOS ONE. After careful consideration, we feel that it has merit but does not fully meet PLOS ONE’s publication criteria as it currently stands. Therefore, we invite you to submit a revised version of the manuscript that addresses the points raised during the review process.

As can be noted from the information below, both reviewers were enthusiastic about this manuscript and the suggested changes are very minor.  

We look forward to receiving your revised manuscript.

Kind regards,

Ruud AW Veldhuizen

Academic Editor

PLOS ONE

Journal Requirements:

Reviewers' comments:

Reviewer's Responses to Questions

**Comments to the Author**

1. Is the manuscript technically sound, and do the data support the conclusions?

Reviewer #1: Yes

Reviewer #2: Yes

2. Has the statistical analysis been performed appropriately and rigorously? 

Reviewer #1: N/A

Reviewer #2: Yes

3. Have the authors made all data underlying the findings in their manuscript fully available?

Reviewer #1: Yes

Reviewer #2: Yes

4. Is the manuscript presented in an intelligible fashion and written in standard English?

Reviewer #1: Yes

Reviewer #2: Yes

5. Review Comments to the Author

Reviewer #1: Dr Vollenweider and colleagues report the results of a systematic review of RCTs investigating the effects of early passive motion of the lower limbs in sedated and mechanically ventilated critically ill patients, on muscle mass preservation and function.

The main findings are 1) that such RCT are scarce, as the authors could find only 5, 2) that those RCTs used different outcome measures, and 3) that they were all small sized, as overall the systematic review involved only 87 patients. The great heterogeneity in methods used for passive motion and in outcome measures across studies precluded the conduct of a metaanalysis.

The authors' conclusion is that the effects of early passive motion in the targeted population are possibly beneficial on a cellular level but that larger size, well conducted randomized studies are needed to 1) confirm the effects of passive motion of the lower limbs, and 2) investigate whether such effects could translate into less muscle wasting during the ICU stay and better muscle function and physical autonomy in the mid- or long-term.

The systematic review is appropriately conducted. The manuscript is well-written and concise. Tables, figures, and Appendices are informative.

I have a few minor comments or suggestions:

-L212, “The Flow Chart in Figure 1 showed the process”. I would write “show” instead of “showed”.

-L317,”… and therefore indicate the importance of early…”. I would temper this assertion because of very small sized studies that were not replicated. I also would add the notion that an estimate of the effect size is very difficult to derive from those studies.

L-356, “…Whether a higher microcirculation can actually positively effect muscles needs…”. It seems it should be “affect”.

Reviewer #2: It is an elegant review properly performed according to systematic review guidelines.

This topic is important but the 5 selected studies, one of these with only 5 patients, makes the results and conclusion weak.

Because of this, I do not agree with the conclusion in the abstract:

'This review indicates early passive motion to benefit on cellular and structural level in sedated and ventilated patients.'

It needs to be changed according to the paper conclusion, that has more caution:

'In conclusion, passive movement show a slight tendency for beneficial changes on cellular level in sedated and ventilated patients in the ICU within the first days of admission, which may indicate a reduction of muscle wasting and could prevent the development of ICU-AW.'

6. PLOS authors have the option to publish the peer review history of their article (what does this mean?). If published, this will include your full peer review and any attached files.

Reviewer #1: No

Reviewer #2: **Yes: **ACGastaldi

---

## [Editor Report · Decision Letter 1]

6 Apr 2022

Passive motion of the lower extremities in sedated and ventilated patients in the ICU – a systematic review of early effects and replicability of Interventions

PONE-D-21-16010R1

Dear Dr. de Bruin,

We’re pleased to inform you that your manuscript has been judged scientifically suitable for publication and will be formally accepted for publication once it meets all outstanding technical requirements.

Kind regards,

Ruud AW Veldhuizen

Academic Editor

PLOS ONE
---

## [Editor Report · Acceptance letter]

20 Apr 2022

PONE-D-21-16010R1 

Passive motion of the lower extremities in sedated and ventilated patients in the ICU – a systematic review of early effects and replicability of Interventions 

Dear Dr. de Bruin:

I'm pleased to inform you that your manuscript has been deemed suitable for publication in PLOS ONE. Congratulations! Your manuscript is now with our production department. 

Kind regards, 

on behalf of

Dr. Ruud AW Veldhuizen 

Academic Editor

PLOS ONE